# Hydroxypropyl-β-Cyclodextrin Depletes Membrane Cholesterol and Inhibits SARS-CoV-2 Entry into HEK293T-ACE^hi^ Cells

**DOI:** 10.3390/pathogens12050647

**Published:** 2023-04-27

**Authors:** Silvia Alboni, Valentina Secco, Bianca Papotti, Antonietta Vilella, Maria Pia Adorni, Francesca Zimetti, Laurent Schaeffer, Fabio Tascedda, Michele Zoli, Pascal Leblanc, Erica Villa

**Affiliations:** 1Department of Life Sciences, University of Modena and Reggio Emilia, 41121 Modena, Italy; silvia.alboni@unimore.it (S.A.);; 2Centre for Neuroscience and Neurotechnology, University of Modena and Reggio Emilia, 41121 Modena, Italy; antonietta.vilella@unimore.it (A.V.);; 3Department of Biomedical, Metabolic and Neural Sciences, University of Modena and Reggio Emilia, 41121 Modena, Italy; 4Department of Food and Drug, University of Parma, 43124 Parma, Italy; bianca.papotti@unipr.it (B.P.);; 5Department of Medicine and Surgery, University of Parma, 43125 Parma, Italy; 6Institut NeuroMyoGène INMG-PGNM Pathophysiologie & Génétique du Neurone et du Muscle, UMR5261, Inserm U1315, 69008 Lyon, France; 7Consorzio Interuniversitario Biotecnologie (CIB), 34148 Trieste, Italy; 8CHIMOMO Department, University of Modena and Reggio Emilia, and Azienda Ospedaliero-Universitaria di Modena, 41124 Modena, Italy

**Keywords:** SARS-CoV-2, cyclodextrin, lipid raft, cholesterol, prophylaxis

## Abstract

Vaccination has drastically decreased mortality due to coronavirus disease 19 (COVID-19), but not the rate of acute respiratory syndrome coronavirus 2 (SARS-CoV-2) infection. Alternative strategies such as inhibition of virus entry by interference with angiotensin-I-converting enzyme 2 (ACE2) receptors could be warranted. Cyclodextrins (CDs) are cyclic oligosaccharides that are able to deplete cholesterol from membrane lipid rafts, causing ACE2 receptors to relocate to areas devoid of lipid rafts. To explore the possibility of reducing SARS-CoV-2 entry, we tested hydroxypropyl-β-cyclodextrin (HPβCD) in a HEK293T-ACE2^hi^ cell line stably overexpressing human ACE2 and Spike-pseudotyped SARS-CoV-2 lentiviral particles. We showed that HPβCD is not toxic to the cells at concentrations up to 5 mM, and that this concentration had no significant effect on cell cycle parameters in any experimental condition tested. Exposure of HEK293T-ACE^hi^ cells to concentrations of HPβCD starting from 2.5 mM to 10 mM showed a concentration-dependent reduction of approximately 50% of the membrane cholesterol content. In addition, incubation of HEK293T-ACE^hi^ cells with HIV-S-CoV-2 pseudotyped particles in the presence of increasing concentrations of HPβCD (from 0.1 to 10 mM) displayed a concentration-dependent effect on SARS-CoV-2 entry efficiency. Significant effects were detected at concentrations at least one order of magnitude lower than the lowest concentration showing toxic effects. These data indicate that HPβCD is a candidate for use as a SARS-CoV-2 prophylactic agent.

## 1. Introduction

Coronavirus disease 19 (COVID-19) is a severe infectious disease caused by a recently discovered coronavirus family member, SARS-CoV-2 (severe acute respiratory syndrome coronavirus 2) [1]. Since its first identification in 2019 in China, it has spread worldwide, resulting in the COVID-19 pandemic. 

Mortality rates were initially very high, but in late 2020, different types of vaccines for COVID-19 became available. Their extensive use has led to a notable change in the clinical scenario. Mortality rates have drastically reduced, and the severity of the clinical picture has generally improved. 

However, despite a full course of vaccination, a considerable proportion of subjects still contract the infection [2]. Furthermore, secondary attack rates for household contacts exposed to fully vaccinated index cases with acute SARS-CoV-2 infection were similar to household contacts exposed to unvaccinated index cases [3]. In addition, approximately 40% of infections in fully vaccinated household contacts arise from fully vaccinated epidemiologically linked index cases [3]. Although breakthrough infections are common, their number in a typical crowded environment is not high. The isolation of infectious virus from the nasopharynx of vaccinated individuals indicates a possible route of infection [4].

This can be further aggravated by the varying infectivity of the various SARS-CoV-2 strains. It has been estimated that each person infected with the Delta strain can infect 3–5 people they come into contact with. However, for the newer strains such Omicron, there is evidence that infectivity is much higher. One person with Omicron could transmit the virus to approximately 20 others [5,6]. Recently isolated Omicron variants such as XBB.1.16, which was associated with the epidemic surge in India, are potentially even more infectious [7]. 

To date, there are no prophylactic measures that can be applied to lower the risk of acquiring the infection for people who, for several reasons, may be at high risk despite vaccination. In this regard, a possible strategy could be to hinder the entry of SARS-CoV-2 into the nasal or pharyngeal mucosa by using a substance that would prevent viral spike protein from combining with membrane receptors. The main receptor for SARS-CoV-2 is the angiotensin-I-converting enzyme 2 (ACE2) receptor, which is present in both the respiratory and digestive systems, with an increasing concentration from the upper to the lower digestive system. SARS-CoV-2 infection initiates with the interaction of the spike protein localized on the virus envelope with the ACE2 receptor. A generic way of interfering with the mechanism of entry of SARS-CoV-2 into cells could exploit the modification of the binding capacity of the ACE2 receptor with the spike protein. The ACE2 receptors are located in microdomains of the cell membrane known as lipid rafts, which rich in cholesterol and sphingolipids. Depletion of cholesterol from the cell membrane through various treatments results in profound modifications, including the relocalization of ACE2 receptors to areas devoid of lipid rafts, leading to a 90% reduction in SARS-CoVinfectivity in cell lines [8]. Cyclodextrins (CDs) are one of the most efficient groups of substances for achieving cholesterol depletion in cell membranes. Among these, hydroxypropyl-β-cyclodextrin (HPβCD), has been extensively studied. Several in vivo and in vitro studies have demonstrated low toxicity of this molecule [9], even when administered intravenously, both in animal models [10] and in humans [11].

In our study, we aimed to verify whether HPβCD could interfere with the entry of SARS-CoV-2 into cells, and whether this could lead to a decrease in its infectivity. To explore this possibility, we used a HEK293T-ACE2^hi^ cell line that stably overexpressed human ACE2 and SARS-CoV-2 Spike-pseudotyped lentiviral particles.

## 2. Materials and Methods

### 2.1. Cells

The human embryonic kidney cells (HEK293T) used in this study were obtained from the ATCC. The HEK293T-ACE^hi^ cell line was engineered to stably overexpress human ACE2 tagged with a hemagglutinin HA flag. Briefly, HEK293T cells were plated at a density of 400,000 cells per well (6-well plate) one day before transfection with 2 μg of pCMV3-ACE2-HA-Hygro encoding construct using the Fugène HD transfection reagent according to the manufacturer’s protocol (Promega, Milan, Italy). One day after transfection, the cells were treated with hygromycin, and selection was completed once non-transfected HEK293T control cells were fully stained.

Cells were cultured in Dulbecco’s modified Eagle medium (DMEM) supplemented with 10% fetal bovine serum (FBS), 2 mM L-Glutamine, and 0.1 mg/mL of penicillin/streptomycin at 37 °C in a humidified atmosphere of 5% CO_2_ in air. All of the reagents used to culture the cells were purchased from Merck KGaA, Darmstadt, Germany. For the experiments, ∼80% confluent cultures were harvested and maintained in complete medium for 24 h. before treatment. 

### 2.2. Drugs

HPβCD was purchased from Wacker Chemie, and α-cyclodextrin (αCD) was purchased from Merck KGaA, Darmstadt, Germany. HPβCD and αCD solutions were freshly prepared and dissolved in HEK293T-ACE^hi^ cell medium (vehicle).

### 2.3. Cytotoxicity Assay

Cell viability was evaluated using the MTT assay to measure cellular metabolic activity as an indicator of cell viability. HEK293T-ACE^hi^ cells were seeded at a concentration of 10^4^ cells/well in 100 µL of complete medium with 10% FBS into 96-well plates. One day post-seeding, a vehicle or HPβCD (concentration range 0.01–40 mM) were added and tested for overnight incubation, followed by a 4 h incubation with a Thiazolyl Blue Tetrazolium Bromide (MTT; Merck KGaA, Darmstadt, Germany) solution (5 mg/mL) at 37 °C in a humidified 5% CO_2_/air atmosphere. The formazan formed was dissolved in 150 µL acid isopropanol (0.1 N HCl in isopropanol) added to all wells. The absorbance was measured by a multiplate reader at a 570 nm wavelength with a 620 nm reference wavelength. All experiments were performed twice (*n =* 8 for each experiment) in independent cultures. The results were expressed as a percentage of the controls.

### 2.4. Cell Cycle Analysis

The analysis of the cell cycle using flow cytometry was performed to investigate the potential cytotoxicity of HPβCD based on the quantification of cellular DNA content using a fluorescent DNA-selective stain, propidium iodide (PI), that exhibits emission signals proportional to DNA mass. HEK293T-ACE^hi^ cells were seeded in a 24-well plate (75,000 cells/well) in a complete medium at 37 °C in 5% CO_2_. After 24 h post-seeding, cells were treated overnight with 5 or 20 mM HPβCD or a vehicle. The cell cycle analysis was carried out immediately after 24 h, or 72 h after the overnight HPβCD treatment to evaluate whether HPβCD can affect the cell cycle stages over time. 

After the treatment or recovery, cells were collected by trypsinization, pelleted by centrifugation for 5 min at 200× *g*, and then resuspended in ice-cold PBS. To permeabilize the samples, 50 μg/mL of RNase A and 50 µg/mL PI staining solution with 0.1% Triton X-100 were added for 20 min at 4 °C in the dark. After incubation, flow cytometry was performed to determine the cell cycle distribution using an Attune NxT Acoustic Focusing Cytometer (Thermo Fisher Scientific, Monza, Italy).

### 2.5. Plasma Membrane Cholesterol Content

Plasma membrane oxidase-sensitive cholesterol was quantified using a radioisotopic assay as previously described [12]. Briefly, HEK293T-ACE2^hi^ cells, 48 h after seeding, were incubated with 3 μCi/mL (1-2,3) of cholesterol (Perkin Elmer, MA, USA) for 24 h in the presence of 2 µg/mL of an acyl-coenzyme A, a cholesterol acyltransferase (ACAT) inhibitor (Sandoz 58035; Merck, Germany), to ensure that all cholesterol was in the free form. Cells were then treated for 18 h with increasing concentrations of HPβCD (from 0.01 mM to 15 mM) or a vehicle, along with 2 µg/mL of ACAT inhibitor. The culture medium was then removed, and cholesterol oxidase enzyme from Streptomyces (Merck, Darmstadt, Germany) dissolved in PBS was added at a concentration of 1 U/mL for 2 h at 37 °C. The PBS was then removed, lipids were extracted from the cell monolayers after 18 h of incubation with 2-propanol, and thin layer chromatography (TLC) was performed to separate radioactive free, esterified, and oxidated cholesterol within the extracted lipid fraction (the mobile phase was composed of 90 mL hexane, 10 mL ethyl ether, and 1 mL methanol). The plasma membrane cholesterol content was expressed as the percentage of oxidated cholesterol over the total cholesterol. In parallel, the total protein content was quantified in cell monolayers using the bicinchoninic acid assay (BCA; Thermo Fisher Scientific, MA, USA) following the manufacturer’s instructions. All experiments were performed twice (*n =* 5–9) in independent cultures.

### 2.6. Plasmids

The FUGW–GFP construct encodes the gene for GFP that has been cloned into the lentivirus plasmid FUGW. This construct was kindly provided by Dr. Xandra O. Breakefield [13]. The psPAX2 plasmid, a second-generation lentiviral packaging construct encoding the HIV-1 Gag, GagPol, Tat, and Rev accessory proteins, was kindly provided by Dr. Didier Trono (EPFL). The pCG1-Spike-HA construct encoding the Wuhan SARS-CoV-2 Spike Envelope glycoprotein was kindly provided by Dr Stefan Pölmann [14] and the pCMV3-2019-nCoV-Spike(S1+S2)-long Flag and untagged (VG40589-CF and VG40589-UT) were purchased from Sino Biological Europe GmbH (Eschborn, Germany).

### 2.7. Pseudotyped Lentiviral Particles Production

Lentivector particles were produced by calcium phosphate DNA transfection of HEK293T cells with the HIV-1 packaging construct pSPAX2, the miniviral genome bearing the expression cassette encoding GFP (FUGW-GFP), and the plasmid encoding either the SARS-CoV-2 Spike Envelope glycoproteins or the VSVg envelope expressing plasmid pMD2.G (addgene). Lentivector particles were also produced without the viral entry glycoprotein as a negative control. For vector production, packaging, transfer, and envelope encoding, plasmids were transfected at a ratio of 8:8:4 μg for 3.5 × 10^6^ cells plated 1 day before transfection in 100 mm dishes. Lentivectors were recovered from the cell culture supernatant 48 h after transfection, centrifuged at 3000× *g* for 5 min, filtrated (0.45 μm filter, Millipore, Burlington, MA, USA), aliquoted, and stored at −80 °C until use.

### 2.8. Single-Cycle Infectivity Assay

HEK293T-ACE^hi^ cells (75,000 cells/well) were seeded in a 24-well plate and cultured in complete medium (w/o antibiotics) at 37 °C in 5% CO_2_ overnight. Twenty-four hours post-seeding, cells (untreated or treated with HPβCD; concentration range from 0.1 to 10 mM) were incubated with 200 μL of pseudotyped viral particles overnight. Over the next three days, cells were washed twice with PBS and maintained in culture with 1 mL of complete medium at 37 °C in 5% CO_2_. After 72 h post-infection, GFP expression was qualitatively evaluated using EVOS Cell Imaging Systems (Thermo Fisher Scientific) and quantitatively evaluated using cytofluorimetry with an Attune NxT Acoustic Focusing Cytometer (Thermo Fisher Scientific) to determine infection efficiency. Three independent experiments were performed (*n =* 5–9).

### 2.9. Curve Fit and Statistical Analysis 

The CD concentrations were converted to Log10 values, and curve fit and IC50 values were calculated using the Nonlinear regression One site—Fit logIC50 function of the Prism software, version 7 (GraphPad Software, San Diego, CA, USA). Statistical significance was assessed by one-way analysis of variance (ANOVA) and post-hoc Dunnett’s test for multiple comparisons against the control (vehicle-exposed) group. A value of *p* < 0.05 was considered statistically significant. All data are presented as mean ± standard error of mean (SEM). Statistical analysis was performed using Prism statistical analysis software, version 7 (GraphPad Software, CA, USA) or SPSS for Windows v.28 (SPSS Inc., Chicago, IL, USA).

## 3. Results

### 3.1. HPβCD Concentration-Dependent Effects on Cell Viability and Cell Cycle

Although HPβCD has been reported to exhibit high biocompatibility, we first evaluated the cytotoxic effects of overnight exposure to HPβCD (concentration range from 0.01 to 40 mM) or a vehicle (Ctrl) on HEK293T-ACE^hi^ cells. The analysis revealed a concentration-dependent effect of HPβCD exposure on cell viability in HEK293T-ACE^hi^ cells (one-way ANOVA: F(9151) = 73.007, *p* < 0.001, followed by post-hoc Dunnett’s test). HPβCD was not found to be toxic up to a concentration of 5 mM (*p* = 0.964). Higher concentrations were increasingly toxic, with a 40 mM concentration of HPβCD resulting in extreme toxicity (reduction of cell viability by ~90%; *p* < 0.001 vs. Ctrl) (Figure 1). 

It has been reported that HPβCD is less toxic than other CDs (including the parent β-cyclodextrins, βCDs) (9). We used the same experimental conditions to evaluate the cytotoxic effects induced by αCD, which is able to extract cholesterol from membranes, although to a lesser extent than HPβCD. We demonstrated that overnight exposure to αCD at concentrations of 2.5 or 5 mM resulting in different effects than those observed for HPβCD (see above). The significant reduction in cell viability was demonstrated through a one-way ANOVA, which showed F(2,23) = 17.974, *p* < 0.001 vs. control, followed by a post-hoc Dunnett’s test (Appendix A).

In leukemic cell lines, HPβCD inhibits cell growth by reducing intracellular cholesterol and inducing G2/M cell-cycle arrest [15], showing that this CD can affect the cell cycle. We then evaluated HPβCD-induced effects on the cell cycle of HEK293T-ACE^hi^ immediately, 24 h, or 72 h after overnight exposure (concentrations: 5 or 20 mM). One-way ANOVA analysis revealed a significant effect on the percentage of cells in the G0/G1 phase (F(2,25) = 16.951, *p* < 0.001) and the G2/M phase (F(2,25) = 3.790, *p* = 0.038) immediately after overnight exposure only at the highest concentration of HPβCD tested (20 mM). This concentration increased the percentage of cells in the G2/M phase (*p* = 0.044, post-hoc Dunnett’s test), while decreasing the percentage of cells in G0/G1 (*p* < 0.001, post-hoc Dunnett’s test). In the same experimental conditions, the proportion of S-phase cells was not significantly affected by HPβCD (Figure 2a). The effect of 20 mM HPβCD on the HEK293T-ACE^hi^ cell cycle was reversible since it had no effect on the HEK293T-ACE^hi^ cell cycle 24 or 72 h after overnight exposure (Figure 2b,c). A concentration of 5 mM of HPβCD had no significant effect on the cell cycle parameters in any experimental condition tested (Figure 2a–c). The percentage of apoptotic cells was not significantly affected by HPβCD treatments at 24 or 72 h or immediately after an overnight exposure (Figure 2).

Overall, our experiments did not show evidence of HPβCD-induced cell toxicity or cell cycle alterations for concentrations up to 5 mM. 

### 3.2. HPβCD Effects on Membrane Cholesterol Content

Cholesterol, and in particular its distribution in membrane rafts, is essential for viral interaction with ACE2 receptors and entry into the cell [16]. Therefore, we evaluated the modulating effect of increasing concentrations of HPβCD with overnight exposure on the membrane cholesterol content of HEK293T-ACE^hi^ cells. The analysis revealed a concentration-dependent reduction of approximately 50% of the initial membrane ^3^H-cholesterol content induced by HPβCD exposure in HEK293T-ACE^hi^ cells (one-way ANOVA: F(7,58) = 18.627; *p* < 0.001, followed by post-hoc Dunnett’s test, Figure 3a). In particular, a significant reduction in membrane ^3^H-cholesterol content was induced by HPβCD exposure starting from 2.5 mM and reaching the maximum inhibitory effect at 10 mM. The concentration-response inhibition curve showed an IC50 value of 1.99 mM (Figure 3b).

### 3.3. HPβCD Effects on Pseudotyped SARS-CoV-2 Particle Entry into HEK293T-ACE2^hi^ Cells

Lentiviral particles that were pseudotyped with SARS-CoV-2 Spike Envelope glycoprotein (HIV-S-CoV2) have been repeatedly used to investigate SARS-CoV-2 entry mechanisms [17,18]. HEK293T-ACE^hi^ cells were incubated overnight with HIV-S-CoV2 pseudotyped particles in the presence of increasing concentrations of HPβCD (from 0.1 to 10 mM) or a vehicle and maintained in culture for 72 h. Three days post-incubation, GFP expression was qualitatively evaluated using EVOS Cell Imaging Systems and quantitatively evaluated using cytofluorimetry to determine the viral particles’ entry efficiency. The analysis revealed a concentration-dependent effect of HPβCD exposure on entry efficiency (one-way ANOVA: F(6,41) = 222.178, *p* < 0.001, followed by post-hoc Dunnett’s test) (Figure 4a–c), with clearly significant effects detected at concentrations at least one order of magnitude lower than the lowest concentration showing toxic effects (see above). The concentration–response inhibition curve showed an IC50 value of 0.78 mM.

As a positive control for viral entry, we performed experiments with HIV-VSVg pseudo-particles. No significant effect on their entry was observed after the modestly cytotoxic 10 mM HPβCD dose (one-way ANOVA: F(2,37) = 21.582, *p* < 0.001, followed by post-hoc Dunnett’s test) (see Appendix A).

## 4. Discussion

The main finding of this study is that HPβCD, at concentrations that are neither cytotoxic nor interfere with cell cycle, markedly reduces (down to approximately 22%) the entry of pseudotyped SARS-CoV-2 particles into HEK293T cells stably overexpressing human ACE2 (HEK293T-ACE^hi^ cells). Pseudotyped HIV-1 lentiviral particles with the SARS-CoV-2 Spike Envelope glycoprotein are widely used to study molecular and cellular mechanisms of SARS-CoV-2 entry [17,18], allowing recognition of the principal membrane receptors and associated proteins necessary for virus entry.

We also demonstrated that HPβCD can decrease membrane ^3^H-cholesterol content by approximately 50%. Remarkably, the IC50 values of the HPβCD-induced decrease in HIV pseudotyped SARS-CoV-2 Spike particle entry and the membrane cholesterol content were close to each other, both at approximately 1 mM. The first effective concentration of HPβCD was close to one order of magnitude below its lowest cytotoxic concentration. Therefore, our hypothesis is that HPβCD-induced decrease in membrane cholesterol content leads to a disruption in membrane organization; more specifically, the cholesterol-rich lipid rafts where many known SARS-CoV-2 receptors are located [16], thus hindering the processes of viral entry into the cell.

Lipid rafts are microdomains of the cell membrane that are rich in cholesterol, sphingolipids, and proteins. This lipid composition causes the membrane to acquire greater rigidity, making it an ideal platform for several receptors including ACE2 [16,19,20]. These structures are critical for coronaviruses, including SARS-CoV-1, to enter into cells and maintain infectivity [8]. Cholesterol depletion in the cell membrane, regardless of the method used, causes profound changes in the membrane’s physical properties [21] and induces the re-localization of receptors localized in the lipid rafts, including ACE2, to areas without lipid drafts [16]. Of further interest is the fact that not only ACE2, but also several other co-receptors that are relevant for SARS-CoV-2 binding and internalization such as heparan sulfate proteoglycan [22], Syndecan-1/4 [23], Neuropilin-1 [24], L-SIGN [25], HDL scavenger receptor B type 1 [26], CD147 [27], and human Toll-like receptors [28] are abundant in the lipid rafts [16]. This means that HPβCD can perturb SARS-CoV-2 binding not only to ACE2 but also to all the other co-localized co-receptors, thus strengthening the inhibitory effect of HPβCD on virus entry.

Another relevant consequence of this putative mechanism of action is that it is variant-independent. This is because the interference with the binding and internalization of SARS-CoV-2 is primarily related to a physical disturbance of the cellular membrane. This is particularly relevant in view of the extremely rapid appearance of SARS-CoV-2 variants with the accumulation of a huge number of mutations that can weaken vaccine protection [29,30].

The motivation for our study was to search for a substance that could be used as a prophylaxis for SARS-CoV-2 infection. This substance would need to be easily delivered to the site of entry of the virus, i.e., the nasal and oropharyngeal mucosa. CDs were identified as ideal candidates due to their longstanding experience in intranasal delivery as carriers of scarcely soluble drugs, showing their safety and ease of use [31,32]. All types of CDs, including αCDs, βCDs, and γ-cyclodextrins (γCDs), can extract cholesterol from cell membranes and disrupt lipid rafts, although with different efficiencies depending on the size and hydrophobicity of their inner cavity [33]. However, αCDs are more efficient in extracting phospholipids than cholesterol [34] and are toxic to cells even at low concentrations, and γCDs are not sufficiently hydrophobic [34,35]. βCDs have the best characteristics to exert this protective action. Among the βCDs, methylβCD (MβCD) and HPβCD are those that have been most studied. Their in vitro activity for cholesterol depletion is similar, as well as their solubility, due to hydrophilic modifications (such as methylation and hydroxypropylation).

Lu et al. were among the first to test MβCD in the context of SARS-CoV infection [8], while Li et al. evaluated the role of MβCD to inhibit SARS-CoV-2 infection in vitro [17]. We used a similar approach, but chose to evaluate HPβCD instead. For the purpose of using CDs as SARS-CoV-2 prophylaxis, MβCD did not meet our requirements. This was because, in the direct comparison between HPβCD and MβCD, the latter was shown to be significantly more toxic both in vitro, even at very low concentrations [36,37], and in vivo [38]. More specifically, the latter study showed that the outer hair cells (OHC) in the organ of Corti appeared histologically normal after treatment with 13 mM of HPβCD, while a higher concentration (27 mM) only caused sporadic damage. Instead, 13 mM of MβCD caused severe damage to the OHC [37]. Notably, in our study, the most effective concentration of HPβCD that was able to inhibit pseudotyped SARS-CoV-2 particle infection (5 mM) was approximately five times lower than the mildly toxic concentration tested in [38]. During the course of our study, a relevant study by Bezerra et al. [39] was published. These authors used experimental models that were partially different from ours (native VERO cells and VERO E6 expressing Transmembrane Protease Serine 2 and Human Angiotensin-Converting Enzyme 2; Calu3 and, as in our study, ACE2 transfected 293T) and showed that HPβCD primarily reduces virus replication and the release of infectious SARS-CoV-2 particles rather than inhibiting entry into the cells. Treatment of human primary monocytes with HPBCD also reduced inflammatory cytokines. Although some relevant experimental conditions were different (e.g., HPβCD was used at much higher concentrations than in the present study), these data are complementary to our findings and indicate an interesting therapeutic potential for this compound.

## 5. Conclusions

In conclusion, data on the ability of HPβCD to interfere with SARS-CoV-2 infection is accumulating, making it a strong candidate as a prophylactic agent.

## Figures and Tables

**Figure 1 pathogens-12-00647-f001:**
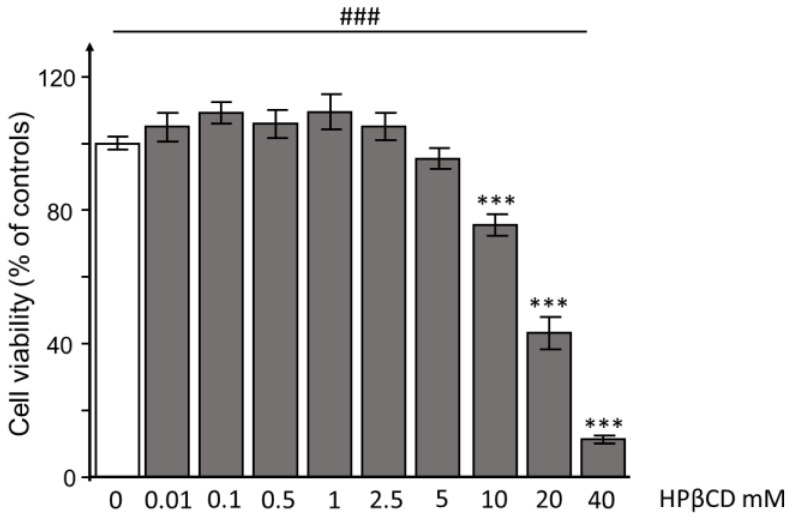
Hydroxypropyl-β-cyclodextrin does not induce cytotoxic effects in HEK293T-ACE^hi^ cells up to a concentration of 5 mM. HEK293T-ACE^hi^ cells were treated overnight with 0.01, 0.1, 0.5, 1, 2.5, 5, 10, 20, or 40 mM of HPβCD (grey columns) or a vehicle (0; white column) and then tested for cell viability (MTT test). Data are the mean ± SEM of two independent experiments (*n =* 8 for each experiment). The concentration effect of HPβCD on cell viability was determined with a one-way ANOVA (### *p* < 0.001) followed by a post-hoc Dunnett’s test vs. control (*** *p* < 0.001). HPβCD: hydroxypropyl-β-cyclodextrin.

**Figure 2 pathogens-12-00647-f002:**
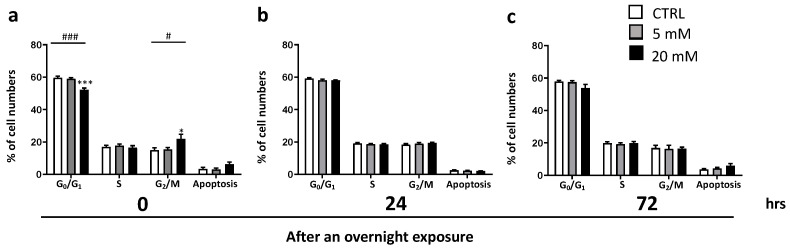
A concentration of 20 mM, but not 5 mM, of hydroxypropyl-β-cyclodextrin transiently affects cell cycle after overnight exposure without significant effect on apoptosis. HEK293T-ACE^hi^ cells were treated overnight with 5 or 20 mM HPβCD or a vehicle (control) and tested for cell cycle distribution analysis using flow cytometry (**a**) immediately after, (**b**) 24 h after, or (**c**) 72 h after HPβCD or a vehicle treatment. Histograms show the percentage of cells in the G0/G1, S and the G2/M phase. Data are presented as the mean ± SEM of two independent experiments (*n =* 5–9). The concentration effect of HPβCD on each cell cycle phase was determined by one-way ANOVA (# *p* < 0.05, ### *p* < 0.001) followed by post-hoc Dunnett’s test vs. controls (* *p* < 0.05, *** *p* < 0.001). HPβCD: hydroxypropyl-β-cyclodextrin. White columns: control; grey columns: 5mM HPβCD; black columns: 20 mM HPβCD.

**Figure 3 pathogens-12-00647-f003:**
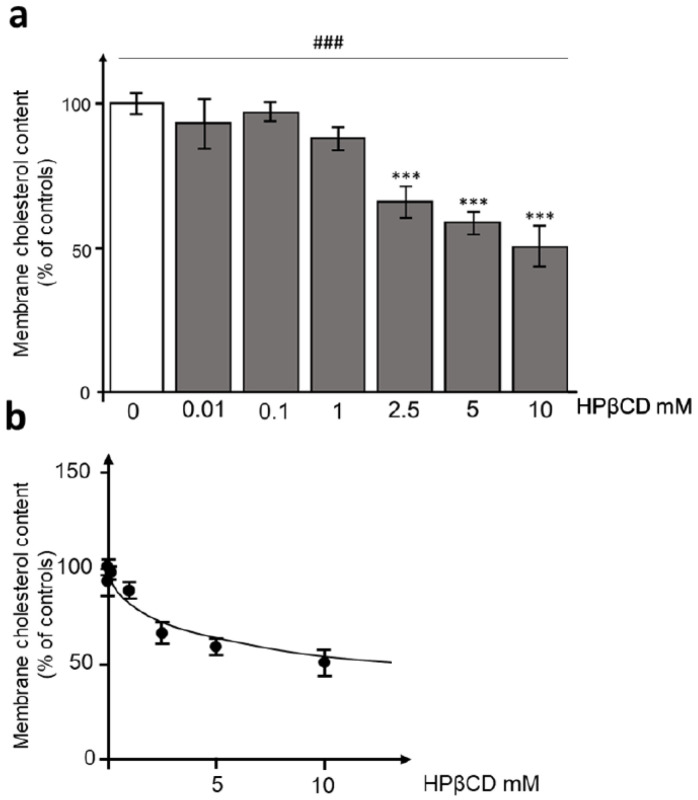
Hydroxypropyl-β-cyclodextrin concentration dependently reduces membrane ^3^H-cholesterol content in HEK293T-ACE^hi^ cells. HEK293T-ACE^hi^ cells were radiolabeled with [3H] cholesterol for 24 h and treated with HPβCD with concentrations ranging from 0.01 to 15 mM (or a vehicle) for an additional 18 h, as described in the Methods. Cholesterol oxidase [1 U/mL] was then incubated for 2 h to oxidize the cholesterol fraction selectively present in cell membranes. Membrane cholesterol content was calculated by counting the radioactivity of oxidized cholesterol over the total cellular radioactive cholesterol. In panel (**a**), the results are reported as the percentage of membrane cholesterol content in HPβCD-treated cells compared to vehicle-treated (control) cells. In panel (**b**), the concentration–response curve was fitted using the Nonlinear regression One site—Fit logIC50 function of the Prism software to obtain the IC50 value. Data are expressed as means ± SEM (*n =* 4 per group). A one-way ANOVA (### *p* < 0.001) followed by the post-hoc Dunnett’s test (*** *p* < 0.001 vs. vehicle-treated cells) was used to analyze the differences between the experimental conditions.

**Figure 4 pathogens-12-00647-f004:**
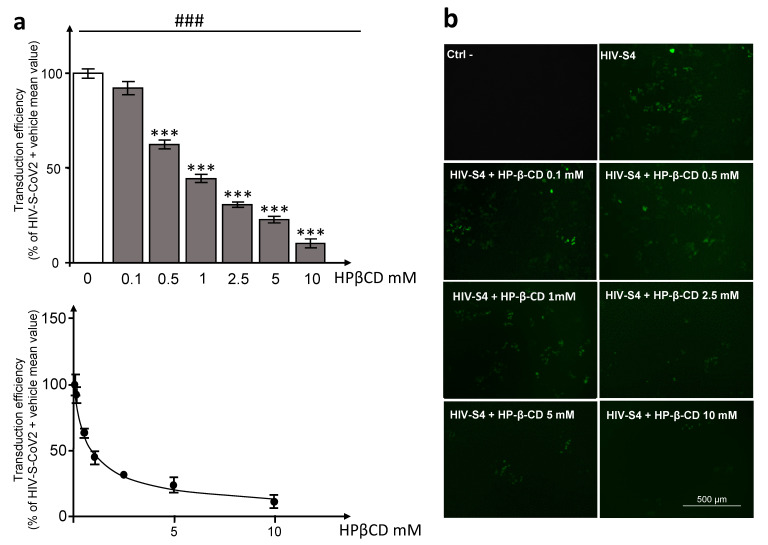
Hydroxypropyl-β-cyclodextrin concentration dependently reduces lentiviral particles pseudotyped with SARS-CoV-2 Spike Envelope glycoprotein entry into HEK293T-ACE^hi^ cells. HEK293T-ACE^hi^ cells were incubated with pseudotyped viral particles containing a GFP expression cassette overnight in the presence of different concentrations (0.1 to 10 mM) of HPβCD or a vehicle. Cells were also incubated overnight with lentivector particles without viral entry glycoprotein containing a GFP expression cassette as a negative control (Ctrl-). Three days post-transduction, (**a**) GFP expression was measured by flow cytometry and (**b**) the concentration–response curve was fitted using the Nonlinear regression One site—Fit logIC50 function of the Prism software to obtain the IC50 value. Qualitative analysis (**c**) was performed using EVOS Cell Imaging Systems Panel on live-HEK293T-ACE^hi^ cells before proceeding to the quantitative analysis. Scale bar = 500 µm. Data are expressed as means ± SEM (*n =* 3–11, three independent experiments). A one-way ANOVA (### *p* < 0.001) followed by the post-hoc Dunnett’s test (*** *p* < 0.001 vs. vehicle treated cells) was used to analyze the differences between the experimental conditions.

## Data Availability

All data generated and analyzed throughout the current study are available from the corresponding author upon reasonable request.

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
