# Peer review of "Hydroxypropyl-β-Cyclodextrin Depletes Membrane Cholesterol and Inhibits SARS-CoV-2 Entry into HEK293T-ACE^hi^ Cells"

_pathogens, 2023, doi:10.3390/pathogens12050647_

Round 1

Reviewer 1 Report

The article tests the compound Hydroxypropyl-β-Cyclodextrin (HPβCD) as a potential inhibitor of SARS-CoV-2 entry into HEK293T-ACE2 cells. For this, the authors first analyze cell viability and cell cycle with increasing concentrations of HPβCD. Subsequently, they analyze the cholesterol content of the membrane of treated cells. Finally, using a viral pseudotype derived from HIV that contains the spike protein, they observe cell entry in the presence of HPβCD.

In my opinion the article has different problems that should be solved:

1.       Introduction: too much emphasis on the transmission of SARS-CoV-2 in a prison. With the background exposed with references 2 and 3 is enough. The introduction could be improved with more background that contributes to the focus of the article.

2.       Only the cytotoxicity assay indicates that two biological replicates were carried out. In the other experiments it is not indicated. It is to be expected that in all of them there are at least three independent biological replicates.

3.       It is not clear what the "Cell cycle analysis" experiment contributes when only 5 or 20 mM of HPβCD are used. Being that in the 20 mM cytotoxicity assay, cell viability significantly decreases.

4.       It is strongly suggested to include a control of cells that constitutively express the ACE-2 receptor, for example Vero-E6 or Huh-7 cells. Since the distribution and/or quantity that the ACE-2 receptor could have in the membrane of HEK293T-ACE2 cells was not analyzed.

5.       It is not understood why all the assays shown use concentrations greater than 5 mM of HPβCD if they significantly decrease cell viability by the MTT assay.

6.       In the Methodology the description 2.8 should be before 2.7

7.       In the production of the viral pseudotype, it is not described how the quantity of particles obtained is quantified. This could be done by ELISA for p24 or by RT activity. Quantification is relevant to compare independent replicates, control pseudotypes or to be able to carry out reproducible assays.

8.       In line 217 Figure A1 does not exist, probably it refers to Figure S1.

9.       The three graphs in Figure 2 do not describe the meaning of each color bar.

1.   To confirm the conclusion, it is strongly suggested to include a positive control of viral entry, such as a pseudotype virus with VSVG.

.   - The resolution of the fluorescent cells' images should be improved and should include DAPI or like for label the cells. It is also suggested to quantify the number of GFP cells (+ or -) by FACS.

Reviewer 2 Report

English should be revised and improved.

The use of radio-isotope to demonstrate cholesterol depletion is not fully accurate. The authors may be only measuring membrane cholesterol movement toward the hydroxy-propyl-cyclodextrin. Unlabelled cholesterol (from the culture medium FCS) may move back to the membrane. Most likely the net "depletion" may be overestimated.

Without a mass measurement , one cannot claim 50% membrane cholesterol depletion. (line 304-305).

I would suggest to comment more on the previous paper (ref 39)  on this cyclodextrin derivative and Covid, to highlight what was missing and what the current manuscript adds to previous knowledge.

Round 2

Reviewer 1 Report

All my observations were corrected or clarified.

Author Response

We thank Reviewer #1 for the comments that helped us to improve our paper.